# Retrieval Competition in Proactive Interference: Effects of Encoding Strength and Consolidation in the Modified Modified Free Recall Paradigm

**DOI:** 10.3390/bs15101332

**Published:** 2025-09-28

**Authors:** Yahui Zhang, Weihai Tang, Xiping Liu

**Affiliations:** 1Faculty of Psychology, Tianjin Normal University, Tianjin 300387, China; zyhmail1218@126.com; 2Development of Psychology, University of Sanya, Sanya 572022, China; twhpsy@126.com

**Keywords:** proactive interference, retrieval competition, Modified Modified Free Recall (MMFR), consolidation, associative dependency, integration and differentiation

## Abstract

This study investigated how encoding strength and consolidation shape proactive interference (PI) in associative memory. Using a Modified Modified Free Recall (MMFR) paradigm, participants studied overlapping (A-B, A-C) and non-overlapping (E-F, G-H) pairs. The encoding strength of List 1 was manipulated (one vs. three study repetitions), while List 2 was held constant. Cued recall was tested immediately and after a 24-h delay. Results showed that increasing List 1’s encoding strength enhanced overall recall for both overlapping and non-overlapping pairs, indicating more effective learning, but did not alter the magnitude of PI. Instead, PI was strongly modulated by retention interval. At immediate test, robust PI emerged across conditions, reflecting cue-based retrieval competition. After a 24-h delay, PI was reduced or absent when List 1 was weakly encoded but persisted in attenuated form when List 1 was strongly encoded, suggesting differential consolidation trajectories for overlapping and non-overlapping associations. Co-retrieval analyses further revealed reliable associative dependency between B and C across all conditions, consistent with representational linkages that promote cooperative retrieval. These findings highlight the dual influence of cue overlap: at the representational level, overlapping pairs form integrated structures that foster co-retrieval, whereas at the retrieval-processing level, cue overload induces competition and PI. Taken together, the results indicate that although initial encoding strength enhances overall recall of List 2, the persistence of proactive interference is influenced by consolidation processes.

## 1. Theoretical Background

In human cognition, the interplay between old and new information often results in memory interference, a key contributor to retrieval failures. Since the early 20th century, interference theory has provided a foundational framework for understanding forgetting, positing that previously or subsequently learned information competes during retrieval ([16]; [23]). Proactive interference (PI)—where prior knowledge (e.g., A-B associations) disrupts the encoding or retrieval of new information (e.g., A-C associations)—and retroactive interference (RI)—where newer information impairs older memories—are the two primary types of interference ([1]). The current study focuses on PI, especially its manifestation within paired-associate learning paradigms, where high cue or content similarity between associations intensifies retrieval competition ([13]).

PI reflects a core limitation of memory: its constrained ability to manage competing traces, particularly when learned materials share semantic or structural similarities. [23] ([23]) demonstrated that recall accuracy declines as the number of previously learned lists increases, highlighting cumulative PI driven by shared retrieval cues ([23]). Conversely, a shift in material category—such as from fruits to professions—can produce a release from PI by reducing competition ([25]). These findings emphasize the dual demands placed on the memory system: maintaining stability (retaining prior traces) while remaining adaptable to new inputs. Thus, PI exemplifies the cognitive challenge of balancing persistence of old memories with acquisition of new information.

### 1.1. Experimental Paradigm and the Role of MMFR

The paired-associate learning paradigm (also known as the A-B, A-C paradigm) is a cornerstone of memory interference research ([12]; [14]), especially for studying similarity-based response competition. In the classic A-B, A-C paradigm, participants learn cue-overlap associate pairs (e.g., A-B followed by A-C), while control participants learn unrelated pairs (e.g., E-F followed by G-H). When recall of A-C is impaired relative to G-H, the resulting deficit is attributed to cue-based interference ([16]). To enhance precision, we adapted this paradigm by treating cue overlap as a within-subject variable, controlling for individual differences while measuring PI through differences in recall performance for A-C versus G-H pairs.

Response competition theory posits that retrieval failures arise when multiple memory traces are simultaneously activated by the same cue, leading to interference ([16]). In standard cued-recall tasks, where participants are required to retrieve only the newer List 2 responses (e.g., A-C), PI manifests as intrusions from older List 1 associations (e.g., A-B), which are often attributed to source monitoring failures ([10]). However, such tasks make it difficult to determine whether the target trace is unavailable or merely inaccessible ([22]). To address this, the Modified Modified Free Recall (MMFR) paradigm ([3]) was developed. By permitting participants to freely recall both B and C for cue A, MMFR reduces output suppression demands and provides a more direct assessment of memory availability

### 1.2. Persistent Competition in MMFR and Its Computational Modeling

While MMFR was initially believed to eliminate retrieval competition by removing the need for selective output ([3]; [9]; [15]), subsequent research challenged this assumption. Cue overload—the activation of multiple associations by a single cue—and differences in associative strength (e.g., stronger encoding of A-C due to recency) suggest that competition persists even in MMFR ([2]; [7]; [13]). Thus, MMFR may reduce—but not abolish—retrieval competition, highlighting the continuing role of memory strength and cue similarity in interference.

The Search of Associative Memory (SAM) model offers a quantitative account of residual competition in MMFR ([18]; [17]). SAM assumes that retrieval involves a sampling process where cues activate memory traces based on their associative strength. A probabilistic selection follows, governed by the ratio rule: P(retrieving item *i*) = strength of *i*/total strength of all candidates. Even in MMFR, competition remains when cue A activates both B and C. When associative strengths are similar, choice ambiguity increases, reducing recall accuracy. In contrast, when one association (e.g., A-B) is much stronger, its dominance reduces competition, improving retrieval efficiency.

In sum, although MMFR reduces inhibitory demands by allowing both responses to be recalled, response competition remains due to cue overload and strength asymmetries, consistent with SAM predictions ([13]). Stronger associations tend to dominate retrieval, delaying or suppressing weaker traces without invoking active inhibition. This non-inhibitory competition explains the persistence of PI in MMFR and underscores the importance of associative strength and cue similarity in shaping memory retrieval dynamics.

### 1.3. Research Questions and Hypotheses

Building on the Search of Associative Memory (SAM) model ([17]; [13]) and the broader response competition framework, the present study addressed two central questions: (a) whether the encoding strength of pre-existing associations (List 1) modulates the magnitude of proactive interference (PI) and (b) whether memory consolidation alters the expression of PI over time.

From a response competition perspective, the degree of interference is determined by the relative strength of target associations (A-C) and competing associations (A-B). When their strengths are comparable, retrieval ambiguity is maximal and PI should be strongest. In contrast, when A-B is substantially stronger than A-C, competition may be dominated by the more accessible association, thereby reducing retrieval ambiguity. If PI scales continuously with the degree of competition, then manipulations of List 1 encoding strength should modulate the magnitude of PI. However, if the presence of competition alone is sufficient to generate PI (a binary relationship), PI should emerge regardless of relative strength, and encoding manipulations would primarily affect overall recall performance without altering the size of the interference effect.

In addition, systems consolidation theory posits that memory traces are reorganized during offline periods, particularly across sleep, through reactivation and redistribution across neural networks ([4]; [20]). Such consolidation may differentially affect overlapping versus non-overlapping associations. If consolidation promotes the integration of overlapping associations ([21]), retrieval of the competitor (B) could enhance access to the target (C), thereby attenuating PI after a 24-h delay. Alternatively, if consolidation exerts no differential influence on overlapping (A-C) versus non-overlapping (G-H) associations, the relative strength of these memories will remain stable across immediate and delayed tests, and consolidation will not modulate the magnitude of PI. Accordingly, the present study tested the following hypotheses:

Encoding strength hypothesis: Increasing List 1 encoding strength will enhance overall recall performance for List 1 pairs. Whether this manipulation modulates PI depends on whether PI is graded by competition strength (reduced PI when A-B is stronger than A-C) or is triggered whenever competition exists (no modulation of PI by encoding strength).

Consolidation hypothesis: A 24-h delay will reduce PI, reflecting consolidation-driven changes in the representational structure of overlapping associations, with stronger benefits for cue-overlapping pairs than for non-overlapping pairs.

Associative dependency hypothesis: We tested whether overlapping associations exhibit cooperative retrieval dynamics under MMFR. Classic views treat retrieval competition as mutually exclusive, such that recalling one trace precludes access to the other. However, because MMFR allows both B and C to be produced for cue A, retrieval may reveal positive associative dependency: recalling one associate increases the likelihood of recalling the other. We therefore hypothesized that overlapping pairs (A-B and A-C) would show reliable co-retrieval dependency across encoding strengths and test delays, reflecting their shared representational structure.

This study aims to elucidate how encoding strength and consolidation jointly shape retrieval competition and PI, refining our understanding of memory dynamics in non-inhibitory recall contexts.

## 2. Methods

The experiments were approved by Tianjin Normal University’s Institutional Review Board. All participants read and signed an informed consent form and were compensated appropriately upon completion. The experiments were approved by the Ethics Committee of the host institution.

### 2.1. Participants

Based on prior research and an a priori power analysis (α = .05, power = .80, two-tailed), a sample size of 52 participants was determined to be sufficient to detect a medium effect size (*f* = 0.25) ([6]). A total of 64 undergraduate students were recruited for the experiment. Data from two participants were excluded due to extremely low recall accuracy, leaving 62 participants (27 males, 35 females) in the final analysis. Participants ranged in age from 17 to 26 years (*M* = 20.21, *SD* = 1.79). Thirty-two participants were assigned to the one-time study condition for List 1, and 30 participants were assigned to the three-time study condition. All participants had normal or corrected-to-normal vision, were tested individually, provided informed consent prior to participation, and received appropriate compensation upon completion of the experiment.

### 2.2. Design

The experiment employed a 2 × 2 × 2 mixed factorial design. List 1 encoding strength, operationalized as the number of study repetitions for List 1 (one vs. three) was manipulated between participants, while pair type (Cue-overlap vs. Non-overlapping) and test time (immediate vs. 24-h-delayed) were manipulated within participants. The dependent variable was memory performance on the Modified Modified Free Recall (MMFR) test for list 2.

### 2.3. Materials

The verbal materials were selected from the Chinese Affective Words System (CAWS) ([24]). Thirty-six verbs and thirty-six nouns were chosen, along with 12 buffer words, resulting in a total of 84 words. The 36 verbs and 36 nouns were randomly assigned to four sets of 18 words each, with each set containing 9 verbs and 9 nouns. To ensure equivalence across sets, the average valence and familiarity ratings were obtained from 32 undergraduate students using a 7-point Likert scale. The mean familiarity ratings for the four sets were 5.590 ± 1.574, 5.653 ± 1.592, 5.658 ± 1.523, and 5.623 ± 1.653, with no significant differences among sets (F(3, 68) = 0.279, *p* = .840). The mean valence ratings were 4.708 ± 0.594, 4.776 ± 0.439, 4.611 ± 0.404, and 4.675 ± 0.385, also showing no significant differences, F(3, 68) = 0.396, *p* = .756.

The image materials were selected from the Bank of Standardized Stimuli (BOSS; [5]). Sixty-six images depicting common objects and animals were chosen, including 12 buffer images. Of the remaining 54 images, 27 depicted familiar living things (e.g., mammals, birds, insects) and 27 depicted familiar objects (e.g., furniture, clothing, buildings, tools). These were randomly and evenly divided into three sets, each containing 9 living things and 9 objects. Familiarity ratings were obtained from 33 participants using a 7-point Likert scale. The mean familiarity ratings for the three sets were 6.401 ± 0.350, 6.465 ± 0.320, and 6.470 ± 0.337, with no significant differences, *F*(2, 51) = 0.236, *p* = .791.

To avoid spurious associations between specific pictures and words, the three picture sets and four word sets were randomly paired to form picture–word associations. In one set, the same pictures appeared in both List 1 and List 2 but were paired with different words to form overlapping A-B and A-C associations. The other two picture sets were paired with two separate word sets to form non-overlap E-F and G-H associations. All picture–word pairs were randomly assigned to different lists and association types.

### 2.4. Procedure

The experiment was programmed using PsychoPy 2024.2.4 and administered on a laptop computer with a 16-inch display (1920 × 1080 resolution; 60 Hz refresh rate). Images were presented in the upper central area of the screen (500 × 500 pixels), and words appeared directly below the images. To minimize primacy and recency effects, the first three and last three trials in the learning phases of both List 1 and List 2 were treated as buffer trials and excluded from statistical analyses. The experimental sequence is illustrated in Figure 1.

**List 1 learning phase.** Each trial began with a fixation cross presented for 500 ms, followed by a picture–word pair. The image appeared in the upper central area of the screen, and the word appeared in the lower central area. Each pair remained on the screen for 5000 ms, during which participants were instructed to memorize the association between the picture and the word. Each list contained 42 pairs, with the first and last three pairs serving as buffers and excluded from analysis. The presentation order of the pairs was fully randomized for each participant to avoid systematic order effects.

**List 2 learning phase**. Following List 1 learning, participants completed a 60-s backward counting task to prevent rehearsal. List 2 learning then proceeded in the same format and timing as List 1. In List 2, half of the pairs shared the same images as in List 1 but were paired with different words, forming overlapping A-B and A-C associations. The other half consisted of pairs with entirely new pictures and words, forming nonoverlapping E-F and G-H associations. To control for potential effects of study order on proactive interference, the sequence of pair presentation in List 2 was also fully randomized across participants. Thus, both List 1 and List 2 associations were presented in random order, ensuring that any observed interference effects could not be attributed to systematic ordering of pair types.

**Cued recall phase.** After List 2 learning, participants completed a 3-min backward counting task before the cued recall test. In this phase, only the image from each studied pair was presented at the center of the screen. Participants were instructed to recall and type the word(s) associated with that image. If an image had been paired with two different words across List 1 and List 2, participants were allowed to provide both responses. The task was self-paced. To prevent potential order effects on proactive interference, the presentation order of cue images was randomized for each participant. Following the immediate cued recall test, participants returned after a 24-h interval to complete a second cued recall test on the same materials, using identical procedures to assess the effects of consolidation on memory performance.

## 3. Result

Data analysis was conducted using R version 4.5.0 ([19]).

### 3.1. Cued Recall Performance for List 2

A 2 (List 1 encoding strength: one vs. three study repetitions) × 2 (pair type: cue-overlap vs. Non-overlap) × 2 (test time: immediate vs. 24-h-delayed) mixed ANOVA on cued-recall accuracy revealed significant main effects of List 1 encoding strength (*F*(1, 60) = 7.11, *p* = .010, generalized *η*^2^ = .057), with higher recall in the three-repetition condition than in the one-repetition condition, and test time (*F*(1, 60) = 28.54, *p* < .001, generalized *η*^2^ = .207), with higher recall at the immediate test than at the delayed test. There was also a significant main effect of pair type (*F*(1, 60) = 18.73, *p* < .001, generalized *η*^2^ = .102), such that non-overlap pairs were recalled more accurately than cue-overlap pairs.

The test time and pair type interaction was significant (*F*(1, 60) = 11.57, *p* = .001, generalized *η*^2^ = .046). Follow-up comparisons showed that the recall advantage for non-overlap over cue-overlap pairs was larger at the immediate test (*M*_difference_ = 0.10, *SE* = 0.017, *t*(60) = 5.76, *p* < .001, Cohen’s *d* = 0.73) than at the delayed test (*M*_difference_ = 0.04, *SE* = 0.019, *t*(60) = 2.06, *p* = .044, Cohen’s *d* = 0.26).

No other interactions reached significance, including the test time and List 1 encoding strength interaction (*F*(1, 60) = 2.70, *p* = .106), the pair type and List 1 encoding strength interaction (*F*(1, 60) = 0.06, *p* = .803), and the three-way interaction (*F*(1, 60) = 2.30, *p* = .135). Means and standard deviations for all experimental conditions are presented in Table 1, and the magnitude of proactive interference (PI) across conditions is illustrated in Figure 2.

### 3.2. Effects of List 1 Encoding Strength and Retrieval Delay on Proactive Interference

To further examine whether cue-overlap elicited proactive interference under different learning and testing conditions, pairwise comparisons were conducted between the cue-overlap and non-overlap pairs within each combination of List 1 encoding strength (one vs. three) and test time (immediate vs. delayed).

In the immediate test condition, participants recalled significantly fewer items in the cue-overlap condition compared to the non-overlap condition, regardless of list 1 encoding strength. When List 1 was encoded once (matching the List 2), recall was significantly lower in the cue-overlap condition (*M* _difference_ = 0.11, *SE* = 0.024, *t*(60) = 4.52, *p* < .001, Cohen’s *d* = 0.58), indicating a robust proactive interference effect. Similarly, when List 1 was encoded three times, recall was also significantly lower in the cue-overlap condition (*M*
_difference_ = 0.09, *SE* = 0.025, *t*(60) = 3.63, *p* = .001, Cohen’s *d* = 0.46).

In contrast, in the delayed test condition, the interference effect was attenuated. When List 1 was encoded once, there was no significant difference in recall between cue-overlap and non-overlap conditions (*M* difference = 0.02, *SE* = 0.027, *t*(60) = 0.84, *p* = .41, Cohen’s *d* = 0.11), indicating that proactive interference was not evident. However, when List 1 was encoded three times, a small but statistically significant interference effect was observed (*M* difference = 0.06, *SE* = 0.028, *t*(60) = 2.06, *p* = .044, Cohen’s *d* = 0.26).

Taken together, these findings demonstrate that proactive interference was most pronounced at immediate test, regardless of encoding strength. After a 24-h delay, interference effects were eliminated when List 1 was weakly encoded but persisted, albeit in attenuated form, when List 1 was strongly encoded. Importantly, the size of the proactive interference effect did not systematically vary as a function of List 1 encoding strength

### 3.3. Recall Performance for A-B Pairs in List 1

To confirm the effectiveness of the encoding-strength manipulation, recall accuracy for A-B pairs was analyzed as a function of study repetitions and test time. As expected, repeated study produced substantially higher recall. At the immediate test, recall was significantly greater in the three-study condition (*M* = .73, *SE* = .04) than in the one-study condition (*M* = .37, *SE* = .04, *t*(60) = −6.67, *p* < .001). At the delayed test, this advantage persisted, with higher recall after three repetitions (*M* = .64, *SE* = .04) than one repetition (*M* = .28, *SE* = .04, *t*(60) = −6.53, *p* < .001).

These results validate the encoding manipulation by demonstrating that three-study A-B associations were retained at a significantly higher level than one-study associations across both immediate and delayed tests. Thus, the manipulation effectively established List 1 associations of different strengths, providing a reliable basis for testing their role as competitors in List 2 recall.

### 3.4. Associative Dependency Analysis

To examine whether the joint retrieval of B and C items exceeded the level expected under independent retrieval, we computed a deviation index for each participant by subtracting the expected joint probability (*P*(B) × *P*(C)) from the observed co-retrieval probability (*P*(B ∩ C)). Positive deviation values indicate that B and C were retrieved together more frequently than predicted under the independence assumption.

To assess whether the joint retrieval of items B and C exceeded the level expected under independent recall, separate one-sample *t* tests were conducted on deviation scores for each combination of encoding strength and test time. Across all conditions, deviation scores were significantly greater than zero, indicating reliable associative dependency between B and C. When List 1 was encoded once and tested immediately, deviation scores were significantly positive (*M* = 0.016, *t*(31) = 2.50, *p* = .018, Cohen’s *d* = 0.44). When List 1 was encoded once and tested after a delay, a similar significant deviation was observed (*M* = 0.022, *t*(31) = 2.95, *p* = .006, Cohen’s *d* = 0.52). For participants who encoded List 1 three times and were tested immediately, the deviation was also significant and larger in magnitude (*M* = 0.026, *t*(29) = 3.65, *p* = .001, Cohen’s *d* = 0.67).

A 2 (List 1 encoding strength: one vs. three study repetitions) × 2 (test time: immediate vs. 24-h-delayed) mixed-design ANOVA on the deviation scores revealed no significant main effect of encoding strength (*F*(1, 60) = 0.55, *p* = .461, generalized *η*^2^ = .008) and no significant main effect of test time (*F*(1, 60) = 0.86, *p* = .358, generalized *η*^2^ = .002). The interaction between encoding strength and test time was also not significant (*F*(1, 60) = 0.82, *p* = .368, generalized *η*^2^ = .002). These results indicate that the degree of dependency between B and C retrieval did not vary as a function of study repetitions or retention interval.

Overall, the findings demonstrate a robust and consistent associative dependency between B and C, with participants recalling the two items together more often than expected under independent retrieval, regardless of encoding strength or retention interval.

## 4. Discussion

The present study investigated how encoding strength of pre-existing associations (A-B) and memory consolidation over time influence proactive interference (PI) in the Modified Modified Free Recall (MMFR) paradigm. By manipulating the number of study repetitions for List 1 items (one vs. three) and assessing List 2 performance at both immediate and 24-h-delayed tests, we aimed to clarify whether relative associative strength or temporal consolidation modulates cue-based competition in memory retrieval.

### 4.1. Proactive Interference and the Role of Test Delay

The results demonstrated that changes in List 1 encoding strength did not significantly alter the magnitude of PI at the immediate test. Robust interference was observed in both one- and three-repetition conditions, indicating that PI is primarily driven by cue overlap and cue diagnosticity, rather than by differences in relative associative strength. This finding suggests that as long as two associations share the same retrieval cue, competition emerges, consistent with classical cue-overload accounts ([23]; [17]).

By contrast, test delay significantly modulated the expression of PI. After a 24-h interval, the interference effect tended to diminish or become attenuated, reflecting differences in the forgetting trajectories of overlapping (A-C) versus non-overlapping (G-H) pairs. Overlapping associations showed relatively better retention over time compared to non-overlapping pairs, which declined more steeply across the delay. This differential forgetting reduced the performance gap. These findings suggest that consolidation processes may differentially stabilize overlapping associations, preserving them more effectively than non-overlapping ones, thereby modulating the degree of proactive interference observed after a delay. Consistently with this view, developmental research has shown a similar pattern: [8] ([8]) reported that a delay eliminated retroactive interference in children, suggesting that consolidation may particularly benefit overlapping associations ([8]).

These forgetting asymmetries can be further interpreted within the framework of memory consolidation. Research on systems consolidation and memory replay during sleep suggests that not all memories benefit equally from offline reactivation; memories with richer associative structures or stronger links to existing networks are more likely to be replayed and stabilized ([4]; [11]). In the present study, overlapping associations (A-C) may have received more replay during consolidation because they were embedded within a richer representational network that included both A-B and A-C links. This greater opportunity for reactivation would have slowed their forgetting relative to non-overlapping pairs (G-H), which lacked associative redundancy and thus benefited less from replay. As a result, A-C associations were relatively better preserved after 24 h than G-H associations in the one-repetition condition, eliminating PI. In the three-repetition condition, however, both A-C and G-H showed similar levels of forgetting, likely because the strong A-B competitor constrained the advantage of A-C replay, allowing residual PI to persist.

### 4.2. Main Effect of List 1 Encoding Strength

In addition to these effects, we observed a robust main effect of List 1 encoding strength: participants who studied List 1 items three times showed higher overall recall not only for A-B pairs but also for List 2 pairs, including both overlapping (A-C) and non-overlapping (G-H) associations. This parallel improvement suggests that repeated practice during List 1 learning enhanced participants’ ability to encode picture–word associations more efficiently, and these improved strategies transferred to subsequent List 2 learning.

Another possible explanation is that with repeated study, List 1 items became more securely encoded, reducing the need to maintain them in working memory. The resulting release of working memory resources may have facilitated the encoding of List 2 associations, producing a general improvement in recall accuracy. Importantly, this effect was nonspecific: it benefited both overlapping and non-overlapping pairs and did not selectively attenuate PI. Thus, while strengthening prior learning can enhance global encoding efficiency, the presence of PI remains primarily determined by cue overlap and diagnosticity.

### 4.3. Co-Existence of Interference and Dependency

Although cue overlap reliably produced proactive interference (lower mean recall for A-C than G-H), overlapping pairs also showed robust positive associative dependency: B and C were co-retrieved more often than expected under independence across all conditions. At first glance, the coexistence of interference and dependency may appear paradoxical. However, this pattern can be explained by distinguishing between representational and retrieval processes.

At the representational level, overlapping associations (A-B and A-C) are stored within interconnected structures, reflecting the integrative property of memory that binds related traces into shared networks. Such structural connectivity fosters cooperative dynamics, allowing activation to spread between items and increasing the likelihood that recalling one associate (e.g., B) supports recall of the other (C). This mechanism explains the stable associative dependency observed across encoding conditions and retention intervals, even when interference varied.

At the retrieval level, however, cue overlap reduces cue diagnosticity by increasing cue overload, forcing the memory system to discriminate among competing traces. Under these conditions, retrieval operates as a competitive decision process: the stronger or more accessible trace tends to dominate, which reduces the marginal recall probability of its competitor and produces proactive interference. Thus, the same structural overlap that promotes co-retrieval at the representational level simultaneously generates retrieval competition at the processing level.

Taken together, these findings highlight the dual influence of cue overlap on memory. Overlap integrates associations into cooperative representational networks, sustaining positive dependency, while at the same time it undermines cue diagnosticity, producing interference during retrieval. The relative balance between these cooperative and competitive effects depends on both encoding conditions and the temporal dynamics of consolidation, which alter forgetting trajectories without disrupting the underlying associative scaffold.

### 4.4. General Summary

In sum, the present study demonstrates three central findings. First, proactive interference (PI) emerged reliably when associations overlapped, but its magnitude was strongly modulated by test delay: PI was robust at immediate test yet attenuated or eliminated after 24 h, reflecting differential forgetting trajectories across overlapping and non-overlapping pairs. Second, increasing the encoding strength of List 1 not only enhanced overall recall for both overlapping and non-overlapping pairs in List 2, but also altered the relative strength between target (A-C) and competitor (A-B) associations at retrieval. Importantly, however, this change in relative strength did not moderate the degree of PI, indicating that interference is determined more by cue overlap and diagnosticity than by the absolute or relative strengths of competing traces. Third, despite the presence of PI, overlapping pairs consistently exhibited positive associative dependency, indicating that shared cues foster both cooperative and competitive dynamics.

Together, these results highlight the dual nature of cue overlap: it promotes interconnected representational structures that support co-retrieval, while simultaneously reducing cue diagnosticity and generating retrieval competition. This layered perspective underscores the need to consider both representational and processing mechanisms in explaining how proactive interference and associative dependency co-occur in memory retrieval.

## 5. Limitations and Future Directions

Several limitations of the present study warrant consideration. First, while the findings demonstrate that proactive interference (PI) and associative dependency can co-exist, the current behavioral measures cannot directly disentangle the underlying mechanisms. Our interpretations in terms of integration and differentiation processes remain inferential, as no direct neural or process-tracing evidence was collected. Future work could incorporate neurocognitive measures such as fMRI pattern similarity, hippocampal activity tracking, or EEG oscillatory markers to provide more precise evidence for how representational structures evolve during consolidation and retrieval.

Second, although the Modified Modified Free Recall (MMFR) paradigm helps reduce output interference, it does not eliminate strategic influences. Allowing participants to recall multiple responses may encourage varied strategies, potentially confounding estimates of associative dependency. Future studies might employ paradigms that more directly separate automatic and controlled retrieval dynamics, such as speeded cued recall, forced-choice recognition, or confidence-based procedures.

Third, the manipulation of encoding strength was limited to study repetition. While repetition reliably increased memory strength, it does not capture other dimensions of encoding quality, such as elaboration, distinctiveness, or associative variability, which may modulate retrieval competition in different ways. Exploring these alternative manipulations would clarify whether the observed pattern of PI and dependency generalizes across different learning conditions.

Finally, the study employed a single 24-h delay, which allowed only a coarse examination of consolidation effects. More fine-grained temporal sampling (e.g., intervals spanning minutes, hours, and multiple days) combined with sleep monitoring could provide richer insights into the temporal unfolding of consolidation. Such approaches would help determine whether integration and differentiation processes occur in parallel or sequentially, and how they interact with cue overload, retrieval practice, and forgetting dynamics across time.

## 6. Conclusions

The present study examined how encoding strength and consolidation jointly shape proactive interference (PI) in a Modified Modified Free Recall (MMFR) paradigm. The findings reveal three key insights. First, increasing the encoding strength of List 1 associations reliably enhanced overall recall performance, not only for List 1 items but also for List 2 pairs, suggesting that prior learning may improve encoding strategies or free up working memory resources. However, this manipulation of relative associative strength did not alter the magnitude of PI, indicating that cue overlap and cue diagnosticity—rather than relative strength—are the primary determinants of interference.

Second, the passage of time significantly modulated PI. After a 24-h delay, interference effects were reduced or eliminated when List 1 was weakly encoded but persisted in attenuated form when List 1 was strongly encoded. This pattern likely reflects differential consolidation of overlapping versus non-overlapping pairs: overlapping associations benefitted more from overnight replay, maintaining their accessibility relative to non-overlapping pairs.

Third, across all conditions, overlapping associations exhibited robust positive associative dependency, with B and C co-retrieved more often than expected by chance. This coexistence of dependency and interference highlights the dual influence of cue overlap. At the representational level, overlapping pairs are stored in interconnected structures that foster cooperative dynamics; at the processing level, however, shared cues reduce diagnosticity and produce retrieval competition.

Taken together, these findings underscore that PI cannot be explained by competition alone. Instead, interference and facilitation coexist in overlapping associative networks, with their relative expression shaped by encoding history and consolidation processes. A comprehensive understanding of PI therefore requires integrative accounts that capture both the cooperative structure of memory representations and the competitive dynamics of retrieval.

## Figures and Tables

**Figure 1 behavsci-15-01332-f001:**
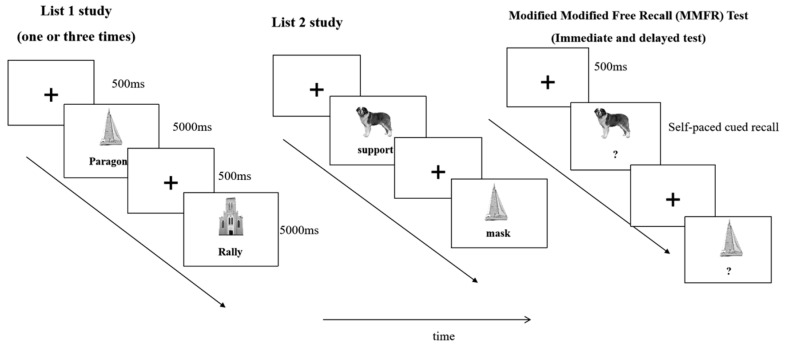
Experimental procedure.

**Figure 2 behavsci-15-01332-f002:**
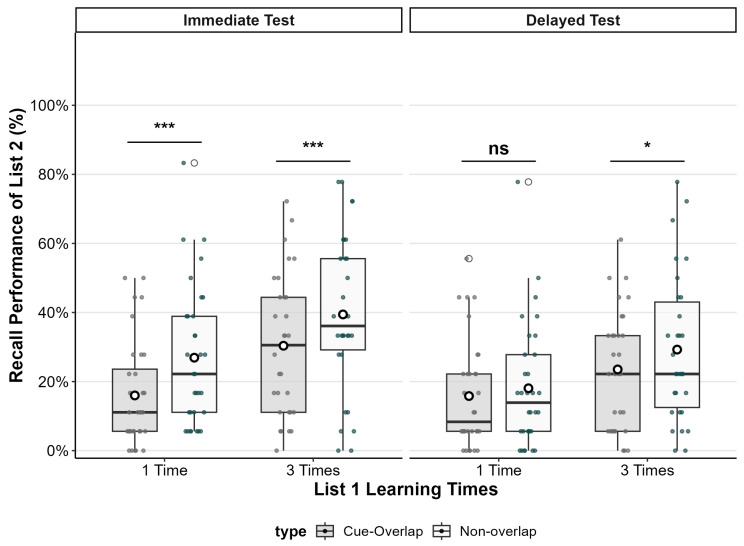
Proactive interference (PI) effects as a function of List 1 learning repetitions and test time. The boxplots display recall performance (%) in list 2 for non-overlap (white boxes) and cue-overlap (gray boxes) pairs in the immediate and delayed tests. Circles represent individual participant scores; horizontal lines indicate medians; white dots within boxes represent condition means. *** *p* < .001, * *p* < .05, ns = nonsignificant.

**Table 1 behavsci-15-01332-t001:** Mean Proportion of Correct Recall for List 2 as a Function of Cue Overlap and Test Delay.

List 1 Encoding Strength	Pair Type	Immediate Test (*IT*)			Delayed Test (*DT*)	
		*N*	*M*	*SD*	*M*	*SD*
One Time	Cue-Overlap (A-C)	32	0.160	0.154	0.158	0.157
	Non-overlap (G-H)	32	0.269	0.202	0.181	0.179
Three Times	Cue-Overlap (A-C)	30	0.304	0.207	0.235	0.174
	Non-overlap (G-H)	30	0.394	0.227	0.293	0.213

## Data Availability

The datasets generated and analyzed during the current study are available from the first author on reasonable request.

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
