# Peer review of "Retrieval Competition in Proactive Interference: Effects of Encoding Strength and Consolidation in the Modified Modified Free Recall Paradigm"

_behavsci, 2025, doi:10.3390/bs15101332_

Round 1
Reviewer 1 Report
Comments and Suggestions for Authors
Zhang and colleagues investigated effects of encoding strength and consolidation on proactive interference (PI) effects using a modified modified free recall (MMFR) design. Participants either encoded a list of word-picture pairs once or three times before learning a second list of pairs. Some of the pairs shared the same picture paired with different words across the two lists (A-B, A-C pairs), whereas others did not (E-F, G-H pairs). Following this, participants underwent an immediate cued-recall test, followed by a delayed cued recall test 24 hours later. The resulted indicated that proactive interference effects (i.e., the difference between overlapping and non-overlapping pairs) were reduced following the delay. The authors suggest that interference was also reduced by presenting the first list of pairs three times compared to just once, but I see no evidence for this claim (see comments below). The authors also used co-retrieval analyses to suggest associative dependency between B and C (which shared the same cue but were never directly paired together). Overall I found the data to be interesting but have significant concerns about the authors’ interpretation of the data, as I explain below in my more specific comments.
First and foremost, the authors make the claim several times, including in the abstract and conclusion sections, that PI was reduced for the participants who studied List 1 three times, but I see no statistical evidence for this at all. PI is typically inferred from reduced recall performance for overlapping compared to non-overlapping pairs, and that is the case here as well (e.g. “We adapted this paradigm by treating cue overlap as a within-subject variable…while measuring PI through differences in recall performance for A-C versus G-H pairs” – lines 59-61). If this is the case, then a reduction of PI due to stronger encoding of A-B pairs would be inferred from a pair type by List 1 encoding strength interaction. But this interaction was not even close to being significant (p=.803). Numerically, in the immediate test, the interference effect was slightly higher in the single encoding condition (M difference = .11) than in the stronger encoding condition (M difference = .09), but the authors don’t report a direct test of the difference, and the difference is so small I can’t imagine that it would be significant, especially with such similar effect sizes (0.58 vs 0.42). In the delayed test, the interference effect was actually only significant in the stronger encoding condition, which would go against the hypothesis of reduced PI due to stronger encoding and reduced cue ambiguity. Despite the lack of evidence for their hypothesis, the authors repeatedly claim that interference was reduced in the stronger encoding condition (e.g., the abstract claims that “Results showed that PI was strongest when both lists were studied once, consistent with maximal cue-based competition. When List 1 was studied three times, PI was attenuated, suggesting reduced retrieval ambiguity.”). I am very concerned about this lack of rigor in the claims that the authors are making in this regard. It should absolutely preclude publication, at least in the manuscript’s current state.
I also have a number of more minor comments, detailed below:
- The most significant minor comment is that performance for non-overlapping pairs clearly benefited in the 3 encoding trials condition, which is what’s keeping them from finding evidence for a reduction of interference in that condition (i.e. since performance improved for both overlapping and non-overlapping pairs). The authors don’t address this effect at all as far as I can tell, but they should. Theoretically, because the E-F and G-H pairs don’t overlap, it seems that there really shouldn’t be any benefit to studying E-F three times on recall of the G-H pairs, unless perhaps participants are forming better associative strategies with more practice with encoding in general or something like that. At any rate the authors should address this.
- I suggest clarifying that the y axis of Figure 2 is recall performance for List 2 pairs only.
- On a very minor note, in the Introduction the authors cite Polyn et al., 2009, when talking about the SAM model, when actually they introduced a different model that is based on retrieved context theory, so I wouldn’t include them in a reference to SAM predictions (line 93).
Reviewer 2 Report
Comments and Suggestions for Authors
The current experiment investigates whether modulating the strength of paired associates can mitigate proactive interference (PI) for subsequently presented overlapping pairs. Using a modified modified free recall (MMFR) paradigm, the authors presented paired associates (A-B) either once or three times, followed by a new list in which some of the items overlapped with those of the prior list (A-C). The authors report that this manipulation did not have a significant effect on the levels of proactive interference observed in the List 2 recalls. Overall, the experiment and analysis are methodologically rigorous and well-executed. My biggest concern has to do with the interpretation of the findings in the Discussion section, which overstates and mischaracterizes the experimental evidence. I provide specific feedback below.
- Did the authors implement any procedures to ensure that overlapping (A–C) and non-overlapping (E–F) pairs were evenly distributed across serial positions in List 2? Serial position effects could interact with interference and should be considered.
- Table 1 reports only List 2 recall performance. However, it would be helpful to also report recall accuracy for List 1 (A–B) items in the overlapping condition. This would clarify the extent to which the initial encoding manipulation succeeded in producing differential baseline strengths.
- The Discussion asserts that PI was stronger when A–B and A–C pairs had equal encoding strength, but the results do not support this claim. In fact, the delayed test suggests the opposite pattern: PI was eliminated in the equal-strength condition but persisted (albeit weakly) in the three-repetition condition. The conclusion that “PI was stronger under equal encoding” should therefore be tempered, as the interaction between encoding strength and test delay was not significant, even within the immediate test condition.
- Similarly, the claim that persistent PI in the 3x study condition after a 24-hour delay reflects stronger competition between the paired associates is not supported by the evidence, given the lack of significant interactions. This interpretation strikes me as an overreach.
- The dual-process integration/differentiation framework proposed in the Discussion is interesting, but the experimental evidence does not clearly support one mechanism over another. The data show some attenuation of PI with delay in the equal-strength condition and a weak residual effect in the strong-encoding condition, but this could be explained in multiple ways. I recommend softening the theoretical claims and more explicitly acknowledging the limits of the current dataset.
Round 2
Reviewer 1 Report
Comments and Suggestions for Authors
I enjoyed the revised version of the paper - it is very much improved and I see no problem with its publication now that the interpretation issues have been addressed. The only thing I would suggest is that you might consider referencing and potentially discussing Darby & Sloutsky, 2015 ("When Delays Improve Memory: Stabilizing Memory in Children May Require Time," published in Psychological Science). That paper reports an elimination of retroactive interference in children following a delay and makes a similar argument about how consolidation might impact overlapping associations more so than non-overlapping. That's just a suggestion and is not necessary for publication.
Author Response
Comments 1 : I enjoyed the revised version of the paper - it is very much improved and I see no problem with its publication now that the interpretation issues have been addressed. The only thing I would suggest is that you might consider referencing and potentially discussing Darby & Sloutsky, 2015 ("When Delays Improve Memory: Stabilizing Memory in Children May Require Time," published in Psychological Science). That paper reports an elimination of retroactive interference in children following a delay and makes a similar argument about how consolidation might impact overlapping associations more so than non-overlapping. That's just a suggestion and is not necessary for publication.
Response1 : Thank you very much for your thoughtful suggestion. I have incorporated the reference you recommended into the Discussion section (page 9, lines 368–371) and added it to the References list (page 13, lines 357–358). I greatly appreciate your insightful feedback, which has helped strengthen the paper.
Reviewer 2 Report
Comments and Suggestions for Authors
I appreciate the authors' thoughtful responses to my comments, and their extensive revisions to the manuscript. My prior concerns have been sufficiently addressed, and I have no further comments.
Author Response
Comments 1 :I appreciate the authors' thoughtful responses to my comments, and their extensive revisions to the manuscript. My prior concerns have been sufficiently addressed, and I have no further comments.
Response1 : We sincerely thank you for your positive feedback and careful evaluation of our revised manuscript. We are grateful that our revisions have adequately addressed your prior concerns, and we greatly appreciate your support for the publication of our work.